

# An ensemble learning-based feature selection algorithm for identification of biomarkers of renal cell carcinoma

Zekun Xin[1,*], Ruhong Lv[2,*], Wei Liu[1], Shenghan Wang[1], Qiang Gao[1], Bao Zhang[1] and Guangyu Sun[3]

[1] Department of Urology, Aerospace Center Hospital, Beijing, China
[2] School of Computer and Information Technology, Beijing Jiaotong University, Beijing, China
[3] Department of Urology, The Second Hospital of Tianjin Medical University, Tianjin, China
[*] These authors contributed equally to this work.

Corresponding authors
Bao Zhang, baozhang_uro@sina.cn
Guangyu Sun, 806860535@qq.com

## ABSTRACT

Feature selection plays a crucial role in classification tasks as part of the data preprocessing process. Effective feature selection can improve the robustness and interpretability of learning algorithms, and accelerate model learning. However, traditional statistical methods for feature selection are no longer practical in the context of high-dimensional data due to the computationally complex. Ensemble learning, a prominent learning method in machine learning, has demonstrated exceptional performance, particularly in classification problems. To address the issue, we propose a three-stage feature selection algorithm framework for high-dimensional data based on ensemble learning (EFS-GINI). Firstly, highly linearly correlated features are eliminated using the Spearman coefficient. Then, a feature selector based on the F-test is employed for the first stage selection. For the second stage, four feature subsets are formed using mutual information (MI), ReliefF, SURF, and SURF* filters in parallel. The third stage involves feature selection using a combinator based on GINI coefficient. Finally, a soft voting approach is proposed to employ for classification, including decision tree, naive Bayes, support vector machine (SVM), k-nearest neighbors (KNN) and random forest classifiers. To demonstrate the effectiveness and efficiency of the proposed algorithm, eight high-dimensional datasets are used and five feature selection methods are employed to compare with our proposed algorithm. Experimental results show that our method effectively enhances the accuracy and speed of feature selection. Moreover, to explore the biological significance of the proposed algorithm, we apply it on the renal cell carcinoma dataset GSE40435 from the Gene Expression Omnibus database. Two feature genes, NOP2 and NSUN5, are selected by our proposed algorithm. They are directly involved in regulating m5c RNA modification, which reveals the biological importance of EFS-GINI. Through bioinformatics analysis, we shows that m5C-related genes play an important role in the occurrence and progression of renal cell carcinoma, and are expected to become an important marker to predict the prognosis of patients.

---

# INTRODUCTION

Feature selection is a crucial process in machine learning and pattern recognition, aimed at selecting a subset of features from a large feature space. The main objectives of feature selection are to enhance the prediction accuracy, eliminate the redundant features, and reduce the time consumption (*Wang, 2011*; *Wang et al., 2022*; *Jiang et al., 2022*). The filter feature selection method is utilized to select a subset of features from high-dimensional data sets without relying on a learning algorithm. While this method is generally quicker, it does not guarantee classifier accuracy. On the other hand, the Wrapper method incorporates a learning algorithm in the classification process to evaluate the accuracy of the selected feature subset. Embedded methods, however, perform feature selection during training and learn the algorithm for the application. Nevertheless, the wrapper and embedded methods incur higher costs compared to the filter method. Due to the limitations of the learning algorithm, their generalization performance is typically lower.

However, those feature selection methods have several main issues. Firstly, most of these methods do not consider redundancy between selected features. Secondly, a single filter-based approach may introduce bias against the selected subset of features. Lastly, inconsistent prediction accuracy can be observed during classification. According to *Rodriguez et al. (2007)*, the performance of classification models can be enhanced by eliminating irrelevant and redundant features from the original dataset. Different feature selection algorithms may select different subsets of features for a given dataset, resulting in varying precision. Therefore, integrating feature selection methods can improve classification accuracy by selecting a stable feature set. When designing integration-based feature selection methods, it is crucial to consider diversity and accuracy (*Wang, Khoshgoftaar & Napolitano, 2012*; *Guo & Zhou, 2019*).

In recent years, machine learning methods have gained widespread attention for feature selection. Ensemble learning, a significant learning approach in machine learning, has demonstrated excellent performance, particularly in classification problems. Ensemble learning involves using multiple base learners to learn and integrate their predictive outputs on input samples (*Cao et al., 2020*; *Hou et al., 2023*). Commonly used ensemble methods include the mean value method, voting method, and learning method. When applied to feature selection, ensemble learning not only integrates multiple models but also obtains various feature subsets. It comprehensively measures features from different perspectives, improving feature availability and effectiveness. It avoids the negative impact of a single result on the model, thereby enhancing model accuracy. Integrated feature selection methods can be classified into homogeneous integration and heterogeneous integration based on the same training data and base learner. Numerous studies have demonstrated that integrated feature selection methods can enhance the model's generalization performance and training speed.

For instance, *Zhou & Guo (2021)* introduced a two-stage feature learning method based on the relative classification information entropy and mutual information entropy, which greatly enhances the efficiency and accuracy of processing high-dimensional data compared to the bagging method. *Li (2018)* presented an integrated feature selection approach that

combines softmax function weighting mechanism, genetic algorithm, and particle swarm algorithm. This study demonstrated the advantages of these three methods in terms of accuracy and efficiency. *Fengshun et al. (2019)* developed a novel CatBoost algorithm by addressing nominal attribute issues within the GBDT framework. They conducted feature selection through IV value analysis, effectively reducing overfitting and achieving promising results in predicting diabetic patients. *Xu & Shen (2021)* proposed a multi-classification detection method for malicious programs based on the XGBoost and Stacking fusion model. The authors utilized Bayesian methods to optimize parameters and employed regularization to overcome overfitting problems. *Wang, Yue & Chen (2018)* put forth a feature selection integration method based on the Analytic Hierarchy Process (AHP). This approach integrates various feature selections into a consistent feature selection process, considering multiple criteria of feature identifiability and independence. The study demonstrated the effectiveness of this method in symbolic data classification. *Kiziloz & Deniz (2020)* introduced a dynamic multi-objective selection model that searches for the optimal set of five classifiers to extract the most representative feature subset. Experimental results on 12 datasets revealed that this method outperforms AdaBoost and Gradient Boosting. *Joodaki, Bagher Dowlatshahi & Joodaki (2022)* proposed an integrated feature selection method based on fuzzy type I-EFSF. This approach applies three different individual feature selection methods to determine feature grades, while using type I fuzzy to handle feature selection uncertainty and reduce noise, thereby improving accuracy, precision, and recall rates. *Miri, Dowlatshahi & Hashemi (2022)* presented an integrated multi-label feature selection method called GMA, based on geometric mean aggregation of text datasets. This method utilizes four different structures of multi-label feature selection algorithms and has demonstrated excellent results on high-dimensional text data. Lastly, *Hoque, Singh & Bhattacharyya (2018)* proposed an integrated feature selection method based on mutual information. This method integrates feature subsets from multiple filter feature selectors and reduces feature redundancy by considering feature-feature and feature-class mutual information. It has achieved impressive results on high-dimensional datasets.

However, most of the datasets used in the aforementioned methods are characterized by low dimensionality. When dealing with high-dimensional data, those feature selection methods often encounter a "dimensional disaster" issue. This arises due to the excessive number of features compared to the limited number of available samples, resulting in poor generalization ability of the feature selection model (*Iffat & Smith, 2009*; *Guyon & Elisseeff, 2003*). Furthermore, it is important to note that an improved feature selection effect is often accompanied by reduced selection efficiency and generalization ability, while enhancing selection efficiency may lead to a loss of precision. Hence, there exists a need to design an algorithm that can achieve high accuracy while maintaining low time and energy consumption when handling high-dimensional data.

To address the problem of feature selection methods for high-dimensional datasets, we propose a three-stage integrated feature selection framework. In the first stage, the Spearman correlation coefficient is utilized for correlation analysis as a data pre-processing step to eliminate highly linear correlated redundant features and reduce the computational time for subsequent feature selection. Then, the f_classif method is employed for the initial

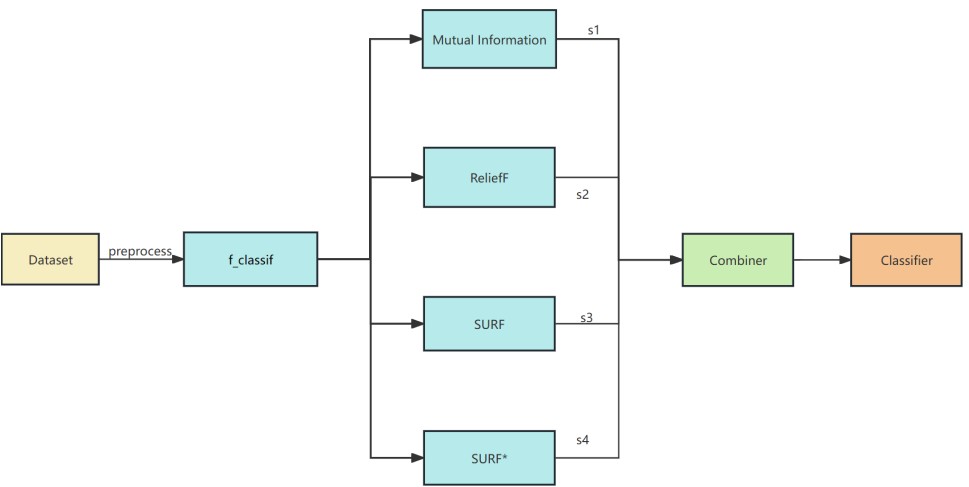

**Figure 1** The overall framework of EFS-GINI.

feature selection, while the MI, ReliefF, SURF, and SURF* filters are processed in parallel to generate four feature subsets for the second stage of selection. Finally, a combinator based on the Gini coefficient is utilized for further screening to obtain the final feature subset. The performance evaluation is conducted using decision tree, naive Bayes, support vector machine (SVM), k-nearest neighbors (KNN), and random forest classifiers through soft voting for classification. Experimental results demonstrate that the proposed method can enhance both the accuracy and efficiency of feature selection for high-dimensional data compared to traditional approaches.

## METHODS

In this section, we propose an integrated feature selection algorithm framework called EFS-GINI and explain each component of the algorithm. This algorithm exhibits strong generality, low complexity, and multiple base selectors to avoid local optimality caused by a single feature selector (*Chandrashekar & Sahin, 2014*; *Khaire & Dhanalakshmi, 2022*).

The proposed algorithm framework is depicted in Figs. 1 and 2. Firstly, data preprocessing is conducted, which involves removing highly linearly correlated features using the Spearman coefficient and standardization. Next, a feature selector based on F-test is employed for the initial stage of feature selection. In the second stage, parallel feature selection is performed using feature filters like MI, ReliefF, SURF, and SURF* to generate four feature subsets. Finally, a combinator based on the GINI coefficient is employed for the third stage of feature selection. The final classifier utilizes decision tree, naive Bayes, SVM, KNN, and random forest algorithms for soft voting in multi-classification and prediction verification.

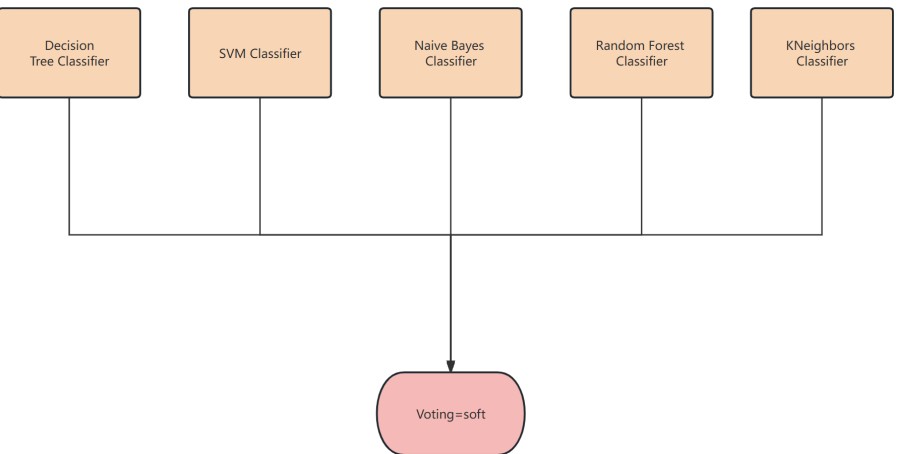

**Figure 2** **The structure of the classifier.**

## Data preprocessing
### *Spearman correlation analysis*

Since the initial dimensionality of the dataset was excessively high, we adopted Spearman correlation analysis, taking into consideration the subsequent feature selection's time complexity. The Spearman rank correlation is a non-parametric test utilized to measure the level of correlation between two variables. The correlation coefficient ranges from −1 to 1. Values closer to 1 indicate a stronger correlation between the variables. Unlike Pearson correlation coefficient, Spearman correlation does not rely on the assumption of continuous level data (intervals or ratios), as it employs grades instead. Additionally, Spearman correlation does not assume a normal distribution for the variables (*Siying, 2019*). The formula for Spearman correlation is as follows:

$$r_s = 1 - \frac{6\sum_{i=1}^{n} d_i^2}{n(n^2 - 1)} \tag{1}$$

where i represents the difference in ranks for each pair of data, and n is the total number of observed samples. In our study, a correlation threshold of 0.9 was set, and features with high correlation were identified by calculating the Spearman correlation matrix.

### *Standardized data*

To mitigate the adverse effects of excessive differences in data dimension levels and enhance model accuracy and convergence speed, this study employs data standardization as a preliminary data preprocessing step, scaling the data to the range $[-1, 1]$. In contrast to data normalization, the standardized operation does not alter the data distribution, rendering it suitable for scenarios where distance-based similarity measurement is applicable. The transformation formula for standardization is as follows:

$$x' = \frac{x - \mu}{\sigma} \tag{2}$$

where $\mu$ represents the mean and $\sigma$ denotes the standard deviation.

## Feature selector based on F-test

An F-test is a statistical method based on analysis of variance that compares variance differences between two or more samples. In the context of feature selection, the F-test is employed to assess the importance of each feature by comparing the variance differences between the response variable and each feature.

The steps involved in conducting an F-test are as follows:

(1)  Calculate the variance (SS) and mean (MS) between each feature and the response variable.
(2)  Compute the variance (SST) and mean (MST) of the population.
(3)  Calculate the F statistic, which is the ratio of SS/MS to SST/MST.
(4)  Determine the critical value of the F statistic using the F distribution table.
(5)  If the computed F statistic exceeds the critical value, it indicates a significant variance difference between the feature and the response variable, thereby identifying the feature as important.

A higher F value or F statistic signifies a larger variance difference between the feature and the response variable, indicating greater importance of the feature in predicting the response variable.

In this study, the feature selector based on the F-test exhibits much faster feature selection times (at the millisecond level) compared to the other four feature selectors (at the second level). Therefore, it is deemed appropriate to utilize the F-test-based feature selector in the initial stage.

## Parallel feature selector

We employ both a ranking-based feature selection method (using Mutual Information) and three search strategy-based feature selection methods (ReliefF, SURF, SURF* (*Urbanowicz et al., 2018*)) to perform parallel feature selection on the preprocessed data.

### Mutual information

Mutual information is a widely used criterion for feature selection in data filtering. In the field of information theory, mutual information I(X;Y) quantifies the uncertainty in X that is reduced by knowing Y. Mathematically, mutual information is defined as follows:

$$I(X;Y) = \sum_{x,y} p(x,y) \log 2 \frac{p(x,y)}{p(x)p(y)}. \tag{3}$$

Here, P(x,y) represents the joint probability distribution function of variables X and Y, while P(x) and P(y) denote the marginal probability distribution functions of X and Y, respectively. We can also express mutual information as:

$$I(X;Y) = H(X) - H(X|Y). \tag{4}$$

Here, H(X) represents the marginal entropy of X, and H(X|Y) represents the conditional entropy of X given Y. Joint entropy, H(X;Y), quantifies the total uncertainty in both X and Y. If H(X) represents the uncertainty of a random variable X, then H(X|Y) measures the amount of uncertainty in X that remains after knowing Y, *i.e.*, the information provided by one variable to another.

Marginal entropy refers to the entropy associated with the marginal distribution of a random variable X. If the marginal distribution is denoted as P(X), the marginal entropy is defined as:

$$H(X) = \sum_i P(x_i) \log 2 \frac{1}{P(x_i)}. \tag{5}$$

In the case of discrete random variables X and Y, the conditional entropy H(Y|X) is defined as follows, considering the joint probability distribution P(x, y) and conditional probability distribution P(y|x):

$$H(Y|X) = -\sum_{x \in X} \sum_{y \in Y} P(x,y) \log 2 P(y|x). \tag{6}$$

### ReliefF

Relief is a feature selection method that is typically used for binary targets. ReliefF, on the other hand, extends Relief to accommodate multiple types of targets, making it a relief-based regression method.

### SURF (Spatially Uiform ReliefF)

The SURF algorithm is a variation of the ReliefF algorithm. In ReliefF, a fixed number of closest neighbors are considered, whereas SURF considers all neighbors within a predefined distance from an individual (*Junwei, 2016*). This distance is known as the similarity threshold T. Thus, SURF selects neighbors that are more similar to the individual than the T threshold. In contrast, ReliefF may utilize a different number of neighbors, potentially overlooking individuals who provide useful information or including individuals who are not informative. Furthermore, SURF incorporates a precomputation of distances, eliminating the need for a user-defined parameter k in the algorithm. This simplification does not compromise the complexity of the algorithm. Additionally, SURF demonstrates a higher success rate in estimating a similarity threshold from the data compared to ReliefF.

### SURF*

The SURF* algorithm is an extension of the SURF algorithm, incorporating the concept of instances that are either closer or farther away from the target. In contrast to SURF, SURF* introduces a T-threshold to determine the proximity of instances. Instances within this threshold are considered close, while those outside are regarded as far away. Notably, SURF* assigns different weights to "far" and "near" cases. Specifically, the difference in eigenvalues for hits is positively weighted (+1), whereas the difference in eigenvalues for misses is negatively weighted (−1). Moreover, we summarize the neighbor selection difference among Relief, ReliefF, SURF, SURF* in Fig. 3.

## Feature combiner based on Gini coefficient

The Gini Index is a measure of feature importance, derived from the Classification and Regression Tree (CART) method (*Lewis, 2000*; *Tangirala, 2020*). In the CART algorithm, feature importance is calculated based on purity enhancement. At each node, the purity

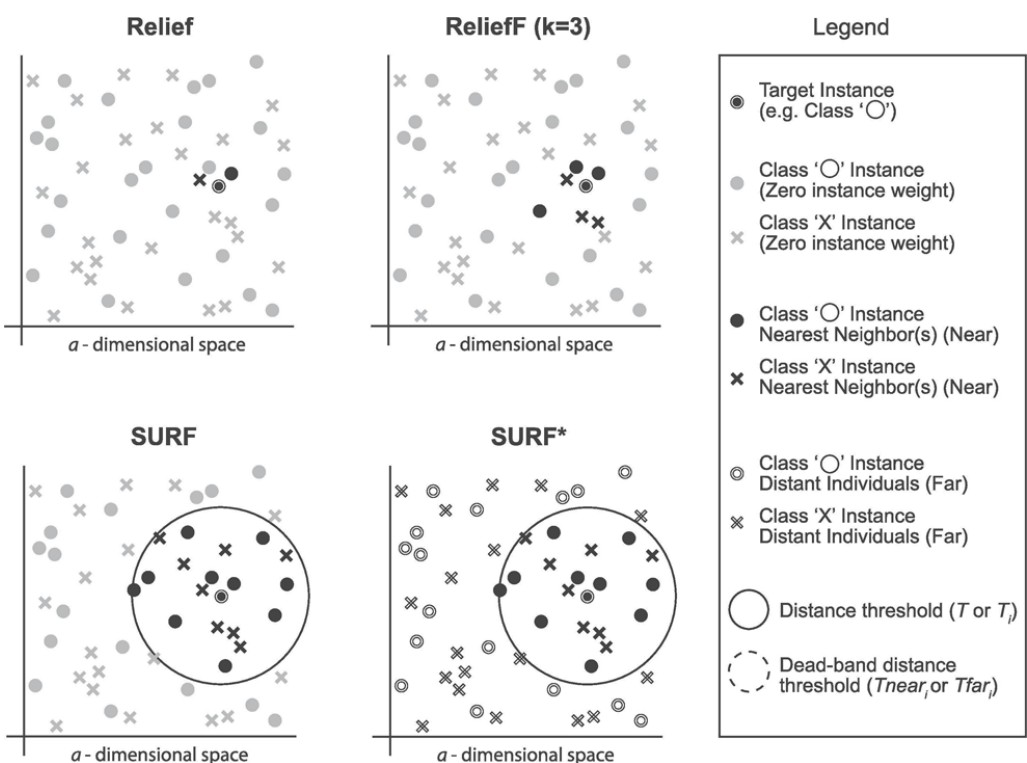

**Figure 3** **The difference of the neighbor selection between Relief, ReliefF, SURF, and SURF\*.**

boost of each feature is calculated, and the feature with the highest boost is selected for node splitting. For classification trees in CART, the GINI value is used for node splitting. A lower GINI value indicates a purer node set and reduces the probability of misclassifying selected samples in the set (*Erhu, 2019*). Specifically, the Gini coefficient for a sample is defined as:

$$Gini(D) = 1 - \sum_{k=1}^{N} P_k^2 \tag{7}$$

where Pk is the proportion of the Kth sample category.

Let's consider a node t, where feature j is chosen for splitting, dividing the dataset D into subsets D1 and D2. The Gini coefficient after splitting with feature j can be calculated as:

$$Gini(D,j) = \frac{|D_1|}{|D|} Gini(D_1) + \frac{|D_2|}{|D|} Gini(D_2). \tag{8}$$

The purity boost of feature j is defined as:

$$Gain(j) = Gini(D) - Gini(D,j). \tag{9}$$

After constructing the decision tree, the importance scores of all features are normalized to have a sum of 1 for comparison. The Gini importance score for feature j is given by:

$$importance(j) = \frac{Gai(j)}{\sum_{j=1}^{m} Gain(j)}. \tag{10}$$

It is important to note that for handling continuous values, the CART and C4.5 algorithms both discretize continuous features. For multi-classification problems, the CART classification tree adopts the idea of continuously dichotomizing discrete features (*Bo, 2018*).

Algorithm 1 provides the complete steps of the feature combinator, which can be divided into the following three steps:

First, add the selected features from subsets S1, S2, S3, and S4 of size k to the feature subset F for final selection. The remaining unselected features form the subset D, selected by the combinator.

Second, construct a CART decision tree and calculate the Gini importance of each feature in D as an evaluation index.

Third, select the k-len(F) features with the highest Gini importance and combine them with F to form the final feature subset.

Algorithm 1 GINI-Select algorithm
Input: feature subsets S1, S2, S3, S4, number of features to be selected k
Output: Optimal feature subset F
1: Initializes the optimal feature collection as $F \leftarrow \varnothing$
2: D ←Initializes the feature subset to be selected as $\varnothing$
3: $j \leftarrow 0$
4: while $j \leq k$ do
5:      ({S1[j]} =S2[j] =S3[j]] =S4[j]}}) then
6:      $F \leftarrow F \bigcup \{S1 [j]$
7: else
8:      $D \leftarrow D\{S_1[j]\}$
9: end if
10: end
11: gini_importance[{D}] =0
12: Build CART decision tree based on data set D
13: Calculate the Gini importance of each feature D[i] in D gini_importance[i]
14: Select the former k-len(D) feature with the greatest Gini importance as f
15: $F \leftarrow F \bigcup f$
16: return optimal subset F

## Soft voting-based classifiers

Voting is a widely used combination strategy in ensemble learning, wherein multiple learners participate in a classification problem and each model's prediction is considered as a single "vote". The final prediction result is determined by the majority vote among the models. In other words, a statistical analysis is conducted on the classification results of K learners, and the class with the highest frequency is chosen as the predicted class. The voting method can be categorized into hard voting and soft voting based on the voting strategies employed. An example of the voting method is shown in Fig. 4.

(1) Hard Voting Mechanism

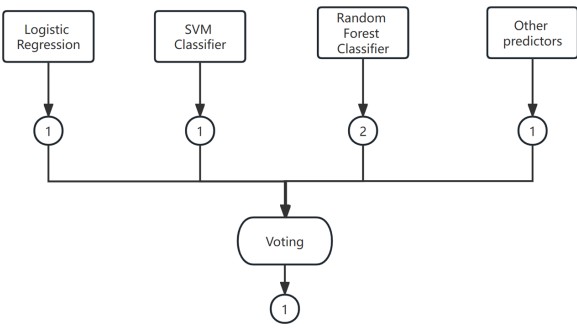

**Figure 4  The voting method.**

A "majority by majority" approach is employed to select the classification results based on the prediction of each model.

(2) Soft voting

The classification rates generated by all classifiers are averaged and selected in this study. The classifier employed in this research is a soft voting classifier, which combines several individual classifiers including decision tree, naive Bayes, support vector machine (SVM), k-nearest neighbors (KNN), and random forest. This ensemble approach effectively mitigates the errors introduced by a single classifier, thereby enhancing the overall classification performance.

# EXPERIMENT AND RESULTS

## Dataset

The data set used in this experiment is a high-dimensional multi-class data set, which are shown in Table 1. The COIL20 dataset is a commonly used image recognition dataset, consisting of 20 objects, each represented by 72 images taken from different angles. Each image in the dataset is converted to a grayscale image of size $128 \times 128$. The ORL dataset is a classic face image dataset containing 400 grayscale face images of 40 individuals, with each person having 10 face images captured in different poses. The images in this dataset have a size of $92 \times 112$ pixels. The warpPIE10P dataset is a face recognition dataset that comprises 4,000 images from 600 individuals, exhibiting different facial expressions, lighting conditions, occlusions, and facial poses. The images in this dataset are of size $32 \times 32$ pixels and have been feature extracted using PCA, resulting in 4,096 features. The Prostate_GE dataset is a gene expression dataset used to predict the tumor grade of prostate cancer (Gleason Score). Each sample in the dataset represents a specific gene expression profile in normal, precancerous, and cancerous tissues. The jaffe dataset is a facial expression database consisting of 213 grayscale images of Japanese women. Each image in the dataset has a size of $256 \times 256$ pixels and exhibits very uniform expressions and postures. The lung dataset is a biological dataset comprising CT images of the lungs along with corresponding annotated data. The TOX_171 dataset is a compound toxicity dataset containing 171 molecules and 12 bioactivity indicators. The Isolet dataset is specifically designed for speech recognition tasks and includes randomized English word sounds. Each

**Table 1** High-dimensional data set used in the experiment.

| Name | Sample number | Number of features | Number of categories |
|---|---|---|---|
| COIL20 | 1440 | 4430 | 20 |
| jaffe | 213 | 676 | 10 |
| lung | 203 | 3312 | 5 |
| ORL | 400 | 1024 | 40 |
| warpPIE10P | 210 | 2420 | 10 |
| Prostate_GE | 102 | 5966 | 2 |
| TOX_171 | 171 | 5748 | 4 |
| Isolet | 1560 | 617 | 26 |

instance in the dataset is represented by 617 features, including 13 linear predictive coding (LPC) coefficients and 13 Mel frequency cepstrum coefficients (MFCC) per frame, as well as fundamental frequency and energy per frame. Detailed information about these datasets can be found at https://jundongl.github.io/scikit-feature/datasets.html.

## Evaluation criteria

In this experiment, we employed various feature selection methods including F-test, MI, ReliefF, SURF, SURF*, and EFS-GINI to conduct a comprehensive control study. Specifically, we selected the top 1% features from the aforementioned datasets.

Accuracy, precision, recall, F1-score (f1_score), and confusion matrix are commonly used evaluation indicators in machine learning. In our study, we set the proportion of the training set and the test set is 60:40 (*Arusada, Putri & Alamsyah, 2017*).

For binary classification, we present the model's prediction confusion matrix in Table 2. In this table, TP (true positive) and TN (true negative) represent the data that was correctly predicted, while FP (false positive) and FN (false negative) represent the data that was incorrectly predicted. TP indicates the correct prediction of a positive example, TN indicates the correct prediction of a negative example, FP indicates the incorrect prediction of a positive example, and FN indicates the incorrect prediction of a negative example. The criteria for accuracy, precision, recall, and F1-score are calculated as follows: accuracy (ACC) is determined by (TP+TN)/(TP+TN+FP+FN), representing the percentage of correct predictions in the total sample; precision (P) is calculated as TP/(TP + FP), indicating the percentage of correctly predicted results in the total sample; recall (R) is TP/(TP +FN), which represents the percentage of correctly predicted results in the total sample. Since precision and recall are conflicting measures, the F1-score (F1) is introduced as 2PR/(P+R) to better evaluate the performance of the representation learner in terms of precision and recall. The closer the F1-score is to 1, the better the classification effect. For the multi-class classification, ACC and F1 can also be calculated using the confusion matrix.

## Experimental environment and parameter settings

The experimental environment of the proposed methods EFS-GINI and the parameter settings in different methods are summarized in Tables 3 and 4 respectively.

**Table 2** Model prediction confounding matrix.

| Real situation | Predicted results | |
|---|---|---|
| | Positive example | Counterexample |
| Positive example | True Positive example TP | False Counter example TP |
| Counter example | False positive example FP | True Counter example TN |

**Table 3** The experimental environment of EFS-GINI.

| Experimental environment | Environment configuration |
|---|---|
| Operating system | Windows 10 64-bit, based on an x64 processor |
| CPU | Intel® Core™ i5-8265U 1.60 GHz |
| RAM | 8G |
| Programming language and version | Python |
| Programming environment | Jupyter Notebook (Anaconda) |

**Table 4** The parameter settings of different methods.

| Parameter in different methods | Setting |
|---|---|
| Threshold of Spearman | 0.9 |
| relifF | n_neighbors = 20 |
| Random forest classifier | n_estimators = 100 |
| SVC | Probability =True |
| precision_score | Average = 'macro' |
| recall_score | Average = 'macro' |
| Voting classifier | Voting = 'soft' |

## Experimental results and analysis

In this section, we conducted performance evaluations on the data set presented in the 'Dataset' section. To ensure a fair evaluation of the models, we adopted the soft voting method for the final classification assessment. Our proposed models were compared with five traditional feature selection methods: mutual information (MI), F test (f_classif), ReliefF, SURF, and SURF*. First, we compared the impact of Spearman's dimensionality reduction. Table 5 displays the number of features before and after dimensionality reduction for different datasets. We focused on evaluating Spearman's optimization of dimensionality reduction using the jaffe dataset as an example. The evaluation metrics used were confusion matrix, accuracy, precision, recall, F1-score, and running time. Figures 5 and 6 depict the confusion matrix before and after dimensionality reduction. The experimental results are presented in Tables 6 and 7, which respectively display the evaluation metrics for different feature selection methods on the jaffe dataset before and after dimensionality reduction. Table 8 shows the improvement in various metrics after dimensionality reduction, comparing the results with and without dimensionality reduction for different feature selection methods. Next, we compared the accuracy, recall rate, and F1-score of the five traditional feature selection methods after Spearman's dimensionality reduction and EFS-GINI on the aforementioned eight datasets. The results

**Table 5  Number of features before and after dimensionality reduction for different datasets.**

| Number of features | jaffe | warpPIE10P | COIL20 | Isolet |
|---|---|---|---|---|
| Before | 676 | 2420 | 1024 | 617 |
| After | 238 | 277 | 119 | 337 |
| Number of features | lung | ORL | TOX_171 | Prostate_GE |
| Before | 3312 | 1024 | 5748 | 5966 |
| After | 3170 | 405 | 5635 | 4818 |

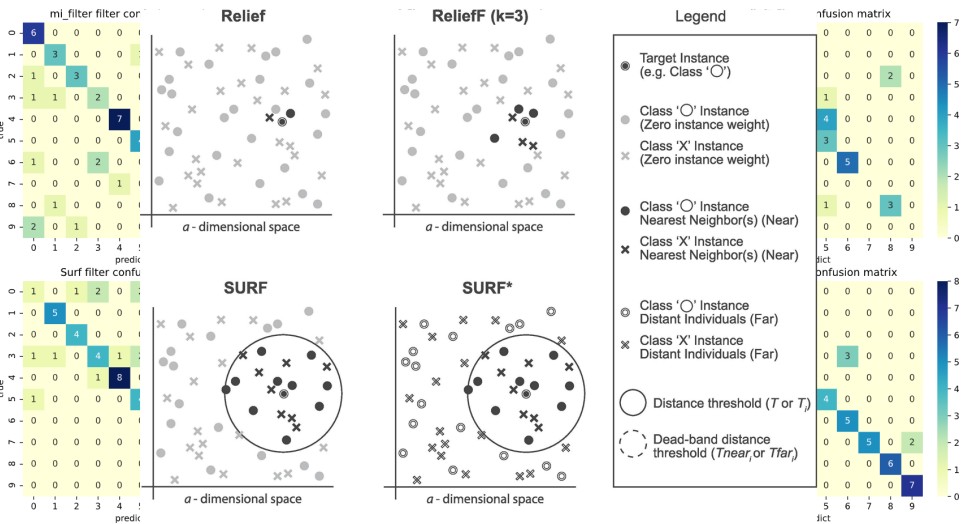

**Figure 5  Confusion matrix of different feature selection methods on the jaffe dataset (before Spearman dimensionality reduction).**

**Table 6  Effects of different feature selection methods on the jaffe dataset (before Spearman dimensionality reduction).**

| Method | Accuracy(%) | Precision(%) | Recall(%) | F1-score | Runtime(s) |
|---|---|---|---|---|---|
| MI | 70.16 | 73.19 | 71.05 | 0.73 | 5.29 |
| F_filter | 60.85 | 65.94 | 57.55 | 0.61 | 0.01 |
| reliefF | 45.16 | 59.70 | 43.26 | 0.53 | 50.94 |
| SURF | 78.68 | 76.87 | 78.09 | 0.77 | 21.08 |
| SURFstar | 73.64 | 75.14 | 73.38 | 0.74 | 47.09 |
| EFS-GINI | 88.37 | 89.01 | 88.50 | 0.89 | 4.81 |

of our experiments on these models are presented in line charts in Figs. 7, 8, 9 and 10. Lastly, we assessed the improvement of EFS-GINI on various metrics compared to the five traditional feature selection methods, as shown in Table 9.

First, we assess the effectiveness of Spearman dimensionality reduction. As shown in Tables 6, 7 and 8, Spearman dimensionality reduction enhances the average accuracy, accuracy rate, recall rate, and F1-score of traditional feature selection methods (F test, mutual information, ReliefF, SURF, SURF*, and ESF-GINI) to varying degrees. Moreover,

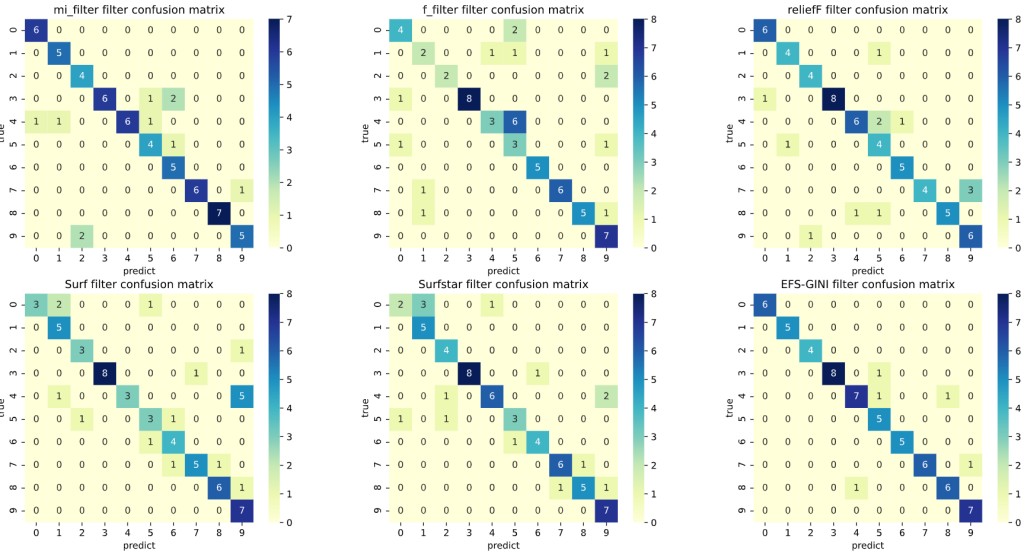

**Figure 6** Confusion matrix of different feature selection methods on jaffe dataset (after Spearman dimensionality reduction).

**Table 7** Effects of different feature selection methods on the jaffe dataset (after Spearman dimensionality reduction).

| Method | Accuracy(%) | Precision(%) | Recall(%) | F1-score | Runtime(s) |
|---|---|---|---|---|---|
| MI | 83.33 | 85.74 | 83.01 | 0.84 | 2.15 |
| F_filter | 73.26 | 78.95 | 73.78 | 0.76 | 0.003 |
| reliefF | 69.77 | 73.88 | 68.66 | 0.71 | 17.56 |
| SURF | 85.66 | 87.47 | 86.09 | 0.87 | 8.99 |
| SURFstar | 79.46 | 82.26 | 81.10 | 0.82 | 18.78 |
| EFS-GINI | 93.02 | 92.9 | 93 | 0.93 | 3.30 |

**Table 8** Enhancement effect of Spearman dimensionality reduction on feature selection.

| Method | Accuracy | Precision | Recall | F1-score | Runtime |
|---|---|---|---|---|---|
| MI | 13.51% | 16.22% | 13.70% | 14.94% | −59.42% |
| F_filter | 19.67% | 19.70% | 29.82% | 24.93% | −62.50% |
| reliefF | 48.94% | 15.63% | 53.33% | 35.14% | −65.53% |
| SURF | 10.26% | 14.29% | 3.85% | 8.85% | −57.35% |
| SURFstar | 8.22% | 9.33% | 10.96% | 10.15% | −60.12% |
| EFS-GINI | 1.10% | 15.97% | 26.65% | 21.30% | −31.50% |

it significantly reduces the running time and improves feature selection efficiency by eliminating linearly dependent redundant features. The impact of Spearman dimensionality reduction on the Jaffe dataset can be observed in the confusion matrices presented in Figs. 5 and 6, where it notably reduces prediction errors.

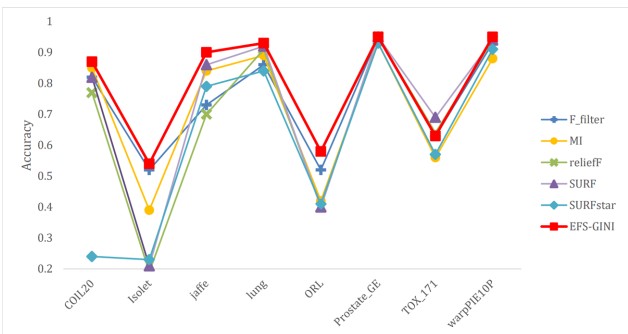

**Figure 7 Line chart of accuracy of different feature selection methods on different datasets.**

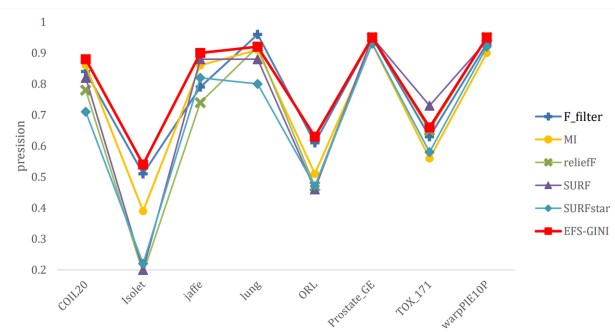

**Figure 8 Line chart of accuracy rate of different feature selection methods on different datasets.**

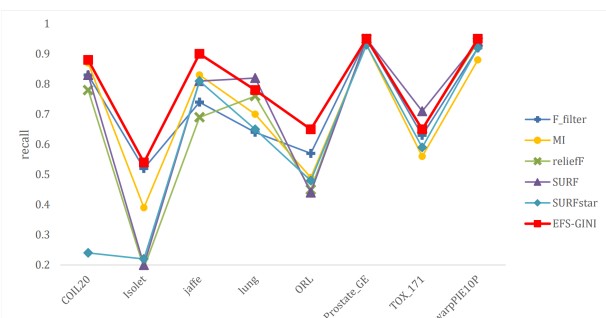

**Figure 9 Line chart of exact rate of different feature selection methods on different datasets.**

Next, we evaluate the effectiveness of EFS-GINI in feature selection. The accuracy, accuracy rate, recall rate, and F1-score line charts demonstrate that EFS-GINI outperforms the other five methods on most datasets. However, it exhibits limited advantages or average performance when dealing with datasets with a small number of samples and classifications, such as lung, Prostate_GE, and TOX_171 datasets. Conversely, EFS-GINI excels in high-dimensional feature selection with a large sample size. An example from the

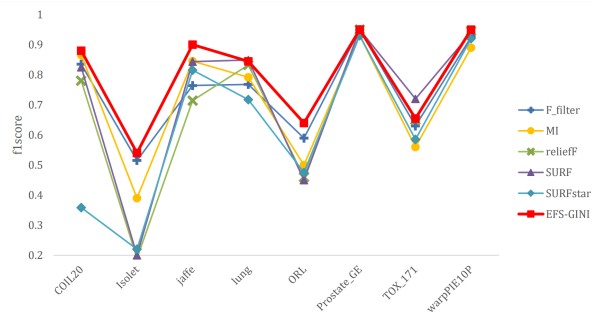

**Figure 10** Line chart of F1-score on different data with different feature selection methods.

**Table 9** Improvement of EFS-GINI compared with traditional feature selection methods.

| Method | Accuracy | Precision | Recall | F1-score | Runtime(s) |
|---|---|---|---|---|---|
| MI | 7.21% | 4.27% | 8.11% | 6.18% | 1.15 |
| F_filter | 21.95% | 13.24% | 21.63% | 17.43% | 3.29 |
| reliefF | 28.05% | 21.01% | 30.70% | 25.85% | −1426 |
| SURF | 4.30% | 2.21% | 4.24% | 3.22% | −5.69 |
| SURFstar | 12.43% | 8.68% | 10.65% | 9.66% | −1548 |

Jaffe dataset shows that EFS-GINI improves accuracy, accuracy rate, recall, and F1-score compared to the five traditional feature selection methods. Furthermore, EFS-GINI exhibits significantly faster running time compared to ReliefF, SURF, and SURF* methods. Thus, the advantages of EFS-GINI in high-dimensional datasets are evident.

## BIOLOGICAL ANALYSIS

To further elucidate the biological significance of the proposed algorithm, we applied it to analyze the renal cell carcinoma dataset GSE40435 obtained from the Gene Expression Omnibus database. By employing our algorithm, we identified a set of genes from GSE40435, which included CLDN10, PROM2, SLC15A4, PRRG2, REEP6, PFKFB4, SLC36A2, FAM151A.1, CAV1.1, SPAG4, NSUN5, and NOP2. Notably, NSUN5 and NOP2 have been implicated in the regulation of m5c RNA modification (*Nombela, Miguel-López & Blanco, 2021*).

It has been documented that mutations in m5C genes are closely linked to a range of human diseases, including nervous system disorders, metabolic diseases, and viral infections (*Barciszewska, 2018*; *Chellamuthu & Gray, 2020*; *Wnuk et al., 2020*). Furthermore, dysregulation of m5C regulators has been observed in various human cancers, such as breast, gallbladder, and bladder cancer (*Haruehanroengra et al., 2020*; *Xue et al., 2020*; *Dong & Cui, 2020*). However, the underlying tumorigenesis mechanism and prognostic implications of dysregulated m5C-related regulators in KIRC remain poorly understood.

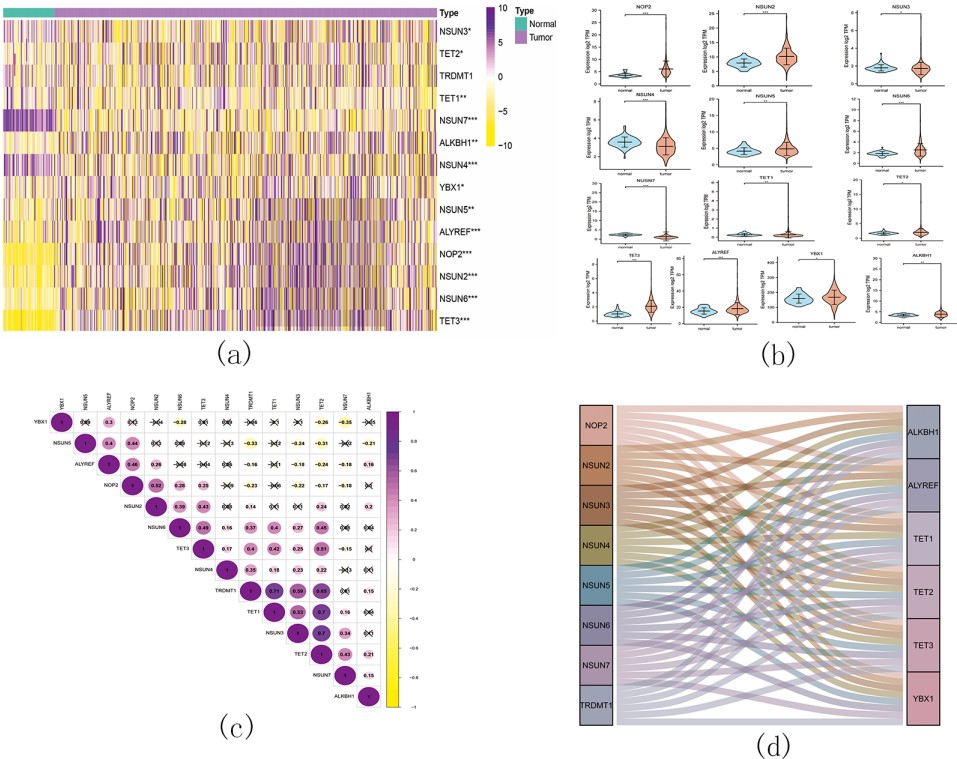

**Figure 11** **The landscape of m5C RNA methylation regulators in KIRC.** (A) Expression levels of 14 m5C RNA methylation regulators in KIRC. The red is upregulated, and the green is downregulated. $^*\ p < 0.05$, $^{**}\ p < 0.01$, $^{***}\ p < 0.001$; (B) The vioplot visualizes the differentially expressed 5C RNA methylation regulators in KIRC. $^*\ p < 0.05$, $^{**}\ p < 0.01$, $^{***}\ p < 0.001$; (C): Spearman correlation analysis of the 14 m5C RNA methylation regulators in KIRC; (D) Sankey of the 14 m5C RNA methylation regulators in KIRC.

In this case study, we utilized The Cancer Genome Atlas (TCGA) datasets to investigate the expression of m5c genes and their correlation with patient prognosis. Additionally, we employed consensus clustering to stratify patients into two distinct clusters with markedly different clinical outcomes. Encouragingly, we discovered that differentially expressed genes within these subgroups were predominantly enriched in immune-related pathways. Lastly, we calculated the risk score for each patient using lasso regression and developed a prognostic risk model for patient survival prediction.

The landscape of m5C RNA methylation regulators in KIRC encompassed a total of 15 related genes. A heatmap analysis clearly demonstrated differential expression of these m5C-related genes in 539 KIRC tissues compared to 72 normal kidney tissues retrieved from the TCGA dataset (Fig. 11A). Specifically, NOP2, NSUN2, NSUN5, NSUN6, TET2, TET3, YBX1, ALKBH1, and ALYREF exhibited significant upregulation in mRNA expression levels, while NSUN3, NSUN4, NSUN7, and TET1 were significantly downregulated in KIRC tissues (Fig. 11B). Furthermore, Figs. 11C and 11D depicts a correlation analysis conducted to gain deeper insights into the intrinsic associations between the 15 m5C RNA modification regulators. This analysis revealed that the correlation between TRDMT1 and TET1 was the most prominent among these regulators.

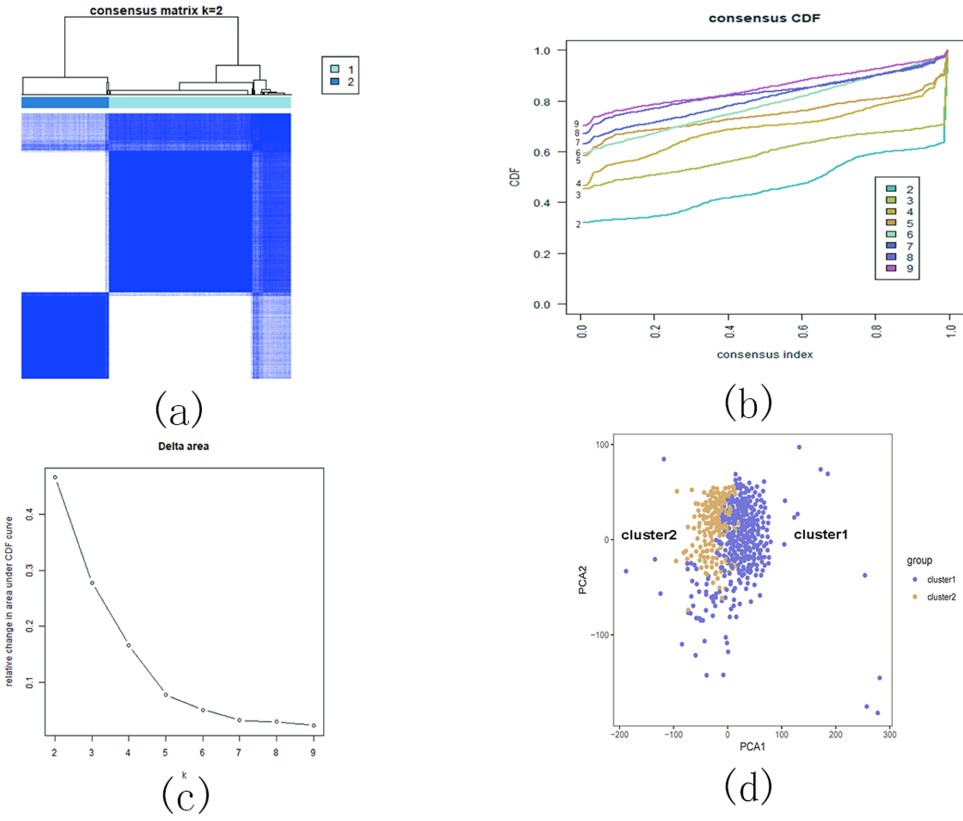

**Figure 12  Consistent cluster analysis of KIRC.** (A) The correlation between subgroups when cluster numbers $k = 2$; (B) Cumulative distribution function (CDF) is displayed for k = 2–9; (C) The relative change in area under the CDF curve for k = 2–9; (D) Principal component analysis of the RNA-seq data.

Based on the similarity of the expression of m5C RNA methylation regulators, we grouped KIRC patients into subgroups. After clustering, $k = 2$ was found to be the optimal number, resulting in two distinct and non-overlapping clusters within the KIRC cohort (Figs. 12A, 12B and 12C). To confirm the validity of our classification, PCA analysis was conducted, revealing that cluster 1 and cluster 2 did not exhibit clear aggregation (Fig. 12D).

Subsequently, we examined the correlation between the identified subgroups and survival rates (OS rates), as well as various clinicopathological characteristics, which included age, gender, stage status, fustat status, T status, M status, and N status. The analysis revealed significant associations between the KIRC subgroups and the OS rates, as well as the clinicopathological features, with the exception of age (Figs. 13A and 13B).

Additionally, the gene ontology (GO) analysis results revealed that the upregulated genes were significantly associated with various malignancy-related processes. These processes include humoral immune response, B cell receptor signaling pathway, humoral immune response mediated by circulating immunoglobulin, and immunoglobulin mediated immune response (Figs. 14A and 14B). The KEGG analysis results revealed significant enrichment of the upregulated genes in various pathways, including viral protein interaction with cytokine and cytokine receptor, TGF-beta signaling pathway,

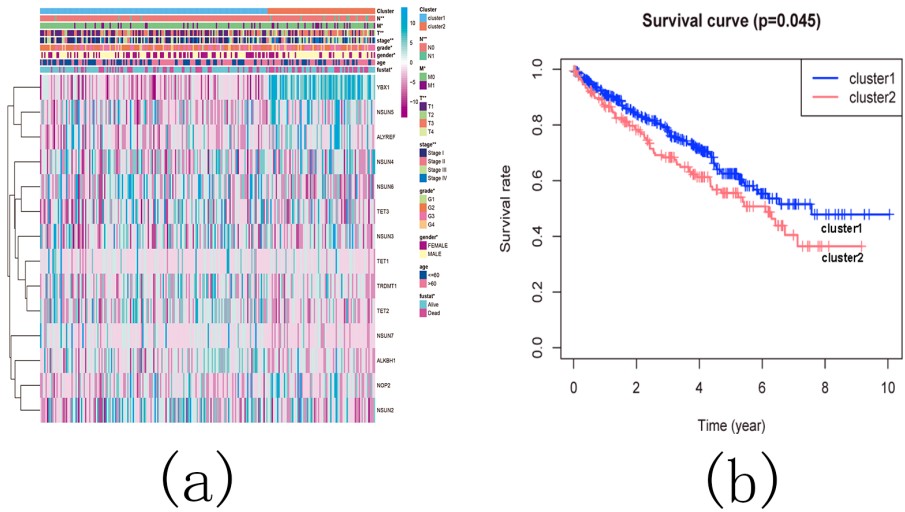

(a)        (b)

**Figure 13** **The difference in clinicopathological features and overall survival between cluster 1 and cluster 2.** (A) Heatmap and clinicopathological characteristics of these two clusters. Green represents low expression and red represents high expression. * $p < 0.05$, ** $p < 0.01$, *** $p < 0.001$; (B) Comparison of overall survival (OS) between cluster 1 and cluster 2.

cytokine-cytokine receptor interaction, and chemokine signaling pathway (Figs. 14C and 14D).

Next, we conducted GSEA analysis on the differentially expressed genes and observed enrichment of genes associated with multiple immune pathways. These pathways include adaptive immune response, adaptive immune response based on somatic recombination of immune receptors utilizing immunoglobulin superfamily domains, humoral immune response, immune effector process, leukocyte-mediated immunity, and lymphocyte-mediated immunity (Fig. 14E).

The estimation analysis revealed significant differences in StromalScore, ImmuneScore, ESTIMATEScore, and TumorPurity between cluster 1 and cluster 2. Specifically, cluster 2 exhibited higher StromalScore, ImmuneScore, and ESTIMATEScore compared to cluster 1, whereas the TumorPurity was lower in cluster 2 as compared to cluster 1 (Figs. 15A and 15B). Furthermore, Spearman's correlation analysis indicated that YbX1, NSUN7, and ESTIMATEScore exhibited the most significant correlation (Fig. 15C).

CIBERSORT analysis revealed that the proportion of 22 different immune cell populations infiltrating cluster 1 and cluster 2 (shown in Fig. 15D). The proportions of CD4 memory activated T cells, regulatory T cells (Tregs), M0 macrophages, and M2 macrophages were significantly higher in cluster 2 compared to cluster 1 (Fig. 15E).

The SSGSEA analysis was also conducted to assess the variance in immune cell infiltration between cluster 1 and cluster 2. There were notable discrepancies observed in the infiltration of 22 distinct immune cell populations, with higher proportions in cluster 2 compared to cluster 1. Some examples include activated B cells, activated CD4 T cells, and activated CD8 T cells (Fig. 15F). These findings suggest that cluster 2 may exhibit a more robust immune response.

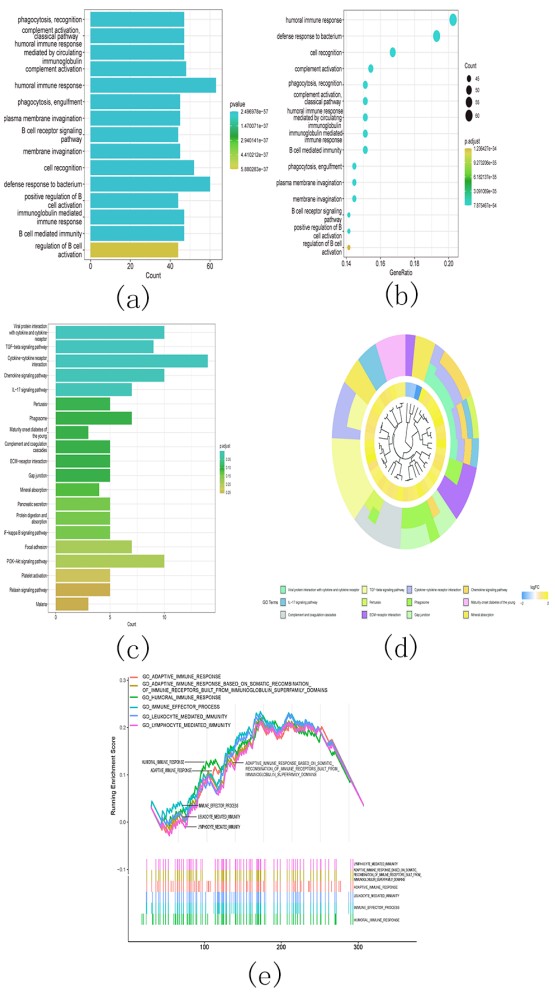

**Figure 14 Gene ontology (GO) analyses, Kyoto Encyclopedia of Genes and Genomes (KEEG) analyses, and Gene Set Enrichment Analysis (GSEA) differentially expressed genes between two clusters.** Function annotation on differently expressed genes in cluster 1 and cluster 2 using GO terms (Figs. 14A–14B), KEGG pathway (Figs. 14C–14D), and GSEA (Fig. 14E). (A) GO analysis; (B) GO analysis; (C) KEGG analysis; (D) KEGG analysis; (E) GSEA analysis.

In order to investigate the prognostic significance of m5C RNA methylation regulators in KIRC, we conducted a univariate Cox regression analysis using the expression levels of these regulators and corresponding clinical survival data. The analysis revealed that NOP2, NSUN2, NSUN5, and NSUN6 were identified as risky genes (HR > 1, $P < 0.05$), while NSUN4, NSUN7, TET2, TRDMT1, and ALKBH1 were identified as protective genes (HR < 1, $p < 0.05$) (Fig. 16A). Additionally, we employed LASSO Cox regression analysis to identify the m5C RNA modification regulators with the strongest prognostic power. Subsequently, six genes (NSUN4, NSUN5, NSUN6, TET2, and ALKBH1) were selected to construct a risk signature for calculating the risk score in KIRC patients (Figs. 16B and 16C). The formula for calculating the risk score is as follows: risk score = 0.686 expression value of NOP2 +0.168 expression value of NSUN5 +0.565 expression value of NSUN6 −

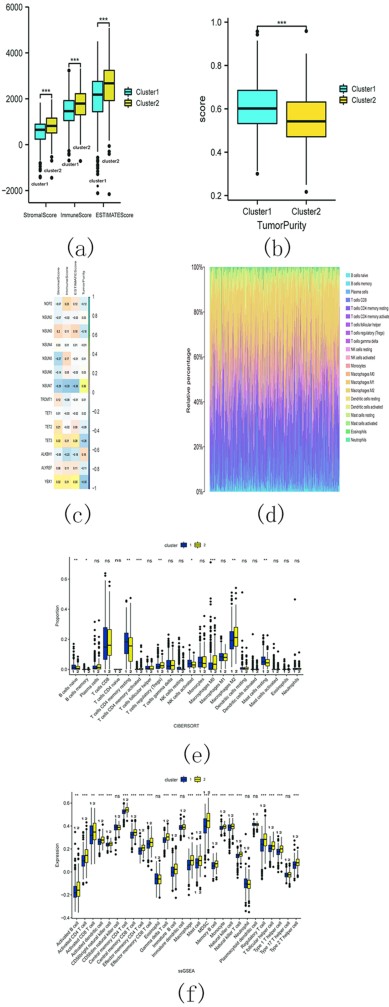

**Figure 15  M5C related genes immune infiltration analysis.** (A) Difference analysis of StromalScore, ImmuneScore, ESTIMATEScore between cluster 1 and cluster 2. (ns, no significance, * $p < 0.05$, ** $p < 0.01$, *** $p < 0.001$); (B) Difference analysis of TumorPurity between cluster 1 and cluster 2. (ns, no significance, * $p < 0.05$, ** $p < 0.01$, *** $p < 0.001$); (C) Association between m5C regulators and StromalScore, ImmuneScore, ESTIMATEScore, TumorPurity; (D) The proportion of 22 kinds of immune cells in tumor tissues; (E) The difference between cluster 1 and 2 through CIBERSORT. (ns, no significance,* $p < 0.05$,** $p < 0.01$, *** $p < 0.001$); (F) Difference analysis of immune cell infiltration in cluster 1 and 2 through ssGSEA. (ns , no significance, * $p < 0.05$, ** $p < 0.01$, *** $p < 0.001$).

0.321 expression value of NSUN4 − 0.463 expression value of TET2 − 0.213 expression value of ALKBH1. Based on the median cut-off value of the risk score, all patients were divided into two groups to establish the risk score model.

Survival analysis demonstrated a significantly worse overall survival (OS) rate in patients with KIRC belonging to the high-risk group (Fig. 17A). A heatmap (Fig. 17B) visualized the expression levels of six prognostic genes in both the high- and low-risk groups. The findings indicated a close correlation between the risk score and several clinical characteristics, including stage, grade, T status, N status, M status, and fustat. Notably,

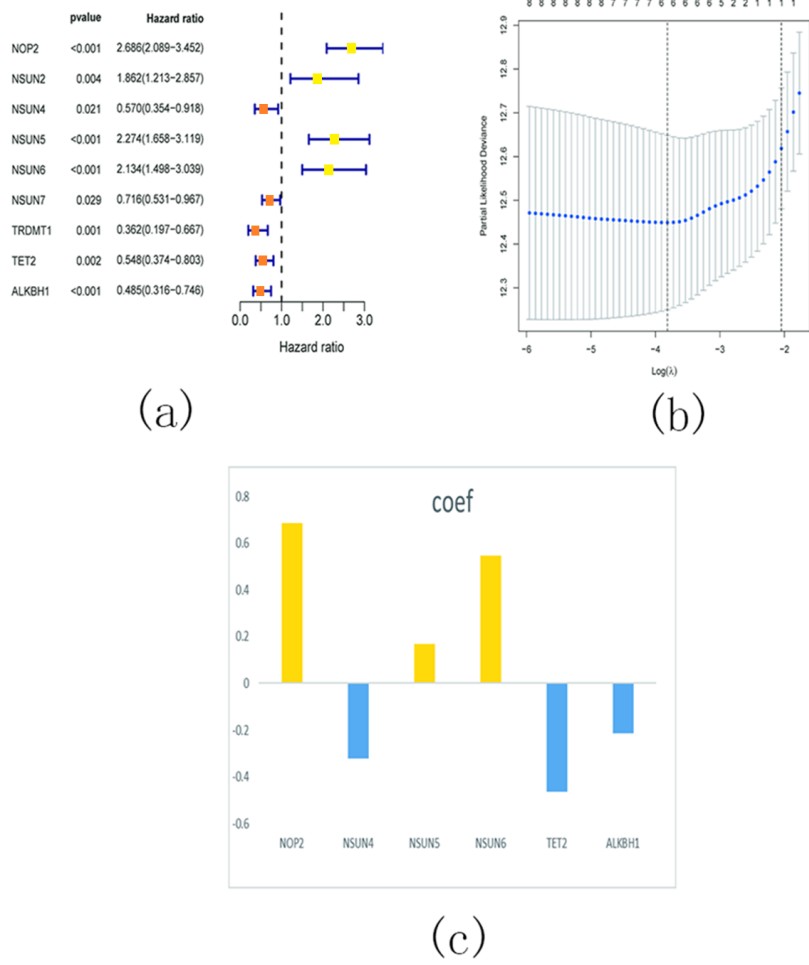

**Figure 16** **Identify a prognostic risk model in KIRC.** (A) Univariate Cox regression analysis of the m5C RNA methylation regulators; (B–C) The coefficients and variable selection using the LASSO model.

the high-risk group tended to have higher T status and N status. To further evaluate the predictive ability of the risk score model, we performed a receiver operating characteristic (ROC) curve analysis. The area under the curve (AUC) values for the 1-, 3-, and 5-year survival predictions were 0.749, 0.719, and 0.712, respectively, indicating a good predictive power for survival outcomes (Fig. 17C).

## CONCLUSION

In this article, a three-stage integrated learning framework, EFS-GINI, is proposed for feature selection in high-dimensional multi-classification datasets. Firstly, we performed data preprocessing which involved standardization and removal of highly linearly correlated features using the Spearman coefficient. Next, we utilized the F test-based feature selector for the initial stage of selection. Additionally, we employed four feature filters, namely MI, ReliefF, SURF, and SURF*, for parallel feature selection, resulting in four feature

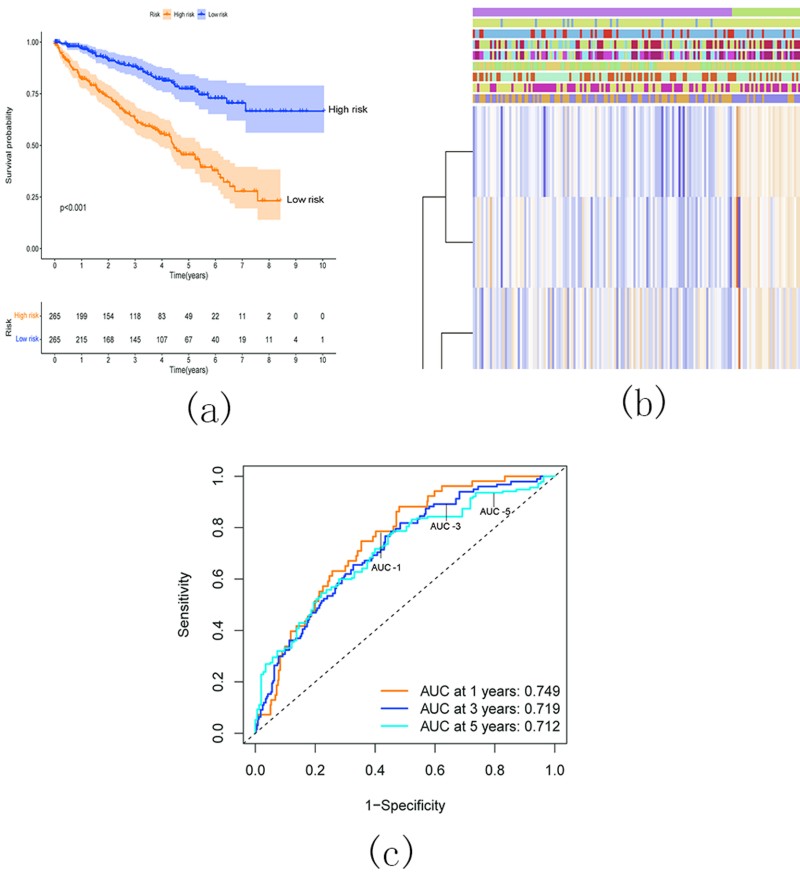

**Figure 17  Relationship between the risk score and the OS, besides clinicopathological features of KIRC.** (A) Kaplan–Meier OS curves for patients in the high- and low-risk groups based on the risk score. The survival probability of the low-risk group is higher than the high-risk group ($p < 0.001$); (B) Relationship between the risk score and the clinicopathological features. The heatmap showed the expression levels of the six m5C RNA methylation regulators in low- and high-risk KIRC patients. It also indicated that the risk score was closely correlated to stage, grade, T status, N status, M status, and fustat of KIRC patients; (C) ROC curves showed the predictive efficiency of the risk signature. The 1-, 3-, and 5-year AUCs were 0.749, 0.719, and 0.712, respectively.

subsets. Furthermore, we applied the Gini coefficient-based combinator for further feature selection. Features selected by all four base selectors were directly added to the feature subset, while features with high Gini coefficients were also included. Finally, a decision tree, naive Bayes, SVM, KNN, and random forest were used for soft voting in the final classifier, facilitating multi-classification and prediction verification. To demonstrate the effectiveness of the proposed algorithm, we conducted experiments on eight high-dimensional datasets, containing a range of 600 to 6,000 features. Experimental results show that our proposed method EFS-GINI effectively exhibits superior performance in high-dimensional multi-classification datasets compared to traditional feature selection methods in terms of accuracy, precision, recall, and F1-score. Moreover, to reveal the biological significance of the proposed algorithm, we apply EFS-GINI on the GSE40435 dataset, the experimental results demonstrate that the gene expression signature of m5c

modification regulators possesses great potential for KIRC prognosis prediction. Our study offers additional evidence for further research regarding m5c RNA modification in KIRC. However, further experimental and clinical exploration are necessary to confirm these finding.

### Funding

The study was funded by the National Natural Science Foundation of China (No. 82302920). The funders had no role in study design, data collection and analysis, decision to publish, or preparation of the manuscript.

### Grant Disclosures

The following grant information was disclosed by the authors:
National Natural Science Foundation of China: 82302920.

### Competing Interests

The authors declare there are no competing interests.

### Author Contributions

- Zekun Xin conceived and designed the experiments, performed the experiments, authored or reviewed drafts of the article, and approved the final draft.
- Ruhong Lv conceived and designed the experiments, authored or reviewed drafts of the article, and approved the final draft.
- Wei Liu conceived and designed the experiments, authored or reviewed drafts of the article, and approved the final draft.
- Shenghan Wang analyzed the data, prepared figures and/or tables, and approved the final draft.
- Qiang Gao analyzed the data, prepared figures and/or tables, and approved the final draft.
- Bao Zhang analyzed the data, prepared figures and/or tables, and approved the final draft.
- Guangyu Sun performed the experiments, authored or reviewed drafts of the article, and approved the final draft.

### Data Availability

The code and data is available at Zenodo:

- LvRuH. (2023). LvRuH/EFS-GINI: EFS-GINI. Zenodo. https://doi.org/10.5281/zenodo.10065391.

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
