# Peer review of "An ensemble learning-based feature selection algorithm for identification of biomarkers of renal cell carcinoma"

_PeerJ Computer Science, doi:10.7717/peerj-cs.1768_

## Round 0.1 · original submission · Major Revisions

As you can see the reviewers have quite a few comments, please take all the comments seriously. For comments that require additional references in the literature, please check carefully if they are really relevant to your paper!

Reviewer 1 ·

Basic reporting

According to the provided information, the paper presents a three-stage feature selection algorithm framework for high-dimensional data based on ensemble feature selection (EFS-GINI). This framework not only demonstrates improvements in feature selection results but also provides in-depth analysis of the biological significance of the proposed algorithm. If the following points can be improved upon, it will enhance the overall organization and logical flow of the article.

Experimental design

The experimental design is detailed and can effectively confirm the viewpoints proposed in the paper.

Validity of the findings

Experimental design can verify the effectiveness of the results

Additional comments

1. In the references section, if that journal or conference is cited only once, the full name should still be used without abbreviation to avoid ambiguity over the source of the abbreviation for readers.

2. I would also suggest to strengthen the bibliography by including the following works that other good general examples of studies on feature selection or ensemble methods, filling a gap in the review of related literature included in the paper.
(1)Wang X, Wang Y, Wong K C, et al. A self-adaptive weighted differential evolution approach for large-scale feature selection[J]. Knowledge-Based Systems, 2022, 235: 107633.
(2)Cao Y, Geddes T A, Yang J Y H, et al. Ensemble deep learning in bioinformatics[J]. Nature Machine Intelligence, 2020, 2(9): 500-508.
(3)Hou Z, Yang Y, Ma Z, et al. Learning the protein language of proteome-wide protein-protein binding sites via explainable ensemble deep learning[J]. Communications Biology, 2023, 6(1): 73.
(4)Jiang L, Sun J, Wang Y, et al. Identifying drug-target interactions via heterogeneous graph attention networks combined with cross-modal similarities[J]. Briefings in Bioinformatics, 2022, 23:2.

3. The author is strongly encouraged to meticulously review and revise the figures and tables to ensure a high level of consistency in font styles, sizes, and other elements throughout the manuscript. Consistency in visual presentation is crucial for enhancing the readability and professional appearance of the document. Therefore, the author should diligently address this aspect to maintain the overall coherence and visual integrity of the research findings.

4. It is imperative that the author pays meticulous attention to the correct usage of English punctuation marks throughout the manuscript. This diligent focus on punctuation details is fundamental in enhancing the clarity, coherence, and overall readability of the paper. By ensuring precise and accurate punctuation, the author can effectively convey their research findings and facilitate a smooth comprehension of the content for readers.

5. The readability and presentation of the study should be further improved. The paper suffers from language problems.

Reviewer 2 ·

Basic reporting

1.The figures should be revised in uniform size for the convenience of readers. And all the formulas should be numbered uniformly and the format of the formulas should be unified to be a format, which seems to be more elegant. For instance, the formula should be left or right aligned or be central.

2.There are too many English grammatical and spelling errors in this paper. The Authors should revise the manuscript carefully and throughly. Take the following sentences for example:
(1).In page 3, line 18, “Feature selection plays a crucial role in classification tasks as part of the data preprocessing process.” should be “Feature selection plays a crucial role in classification tasks as a part of the data preprocessing process.”

(2).In page 5, line 46, “The main objectives of feature selection are to enhance prediction accuracy, eliminate redundant features, and reduce time consumption during analysis. [1]” should be “The main objectives of feature selection are to enhance the prediction accuracy, eliminate the redundant features, and reduce the time consumption [1].“

(3).In page 15, line 403, “Conversely, EFS-GINI excels in processing high-dimensional classifications with a large sample size.” should be “Conversely, EFS-GINI excels in high-dimensional feature selection with a large sample size.”

(4).In page 16, line 431. “were significantly decreased in KIRC tissues(Figure 11B),In addition, as shown in Figure 11 (C-D)” should be “were significantly decreased in KIRC tissues (Figure 11B). In addition, as shown in Figure 11 (C-D)”.

(5).In page 16, line 470. “the StromalScore, ImmuneScore, ESTIMATEScore of cluster 2 was higher than that of cluster 1” should be “the StromalScore, ImmuneScore, ESTIMATEScore of cluster 2 were higher than that of cluster 1”

Experimental design

1. What’s the running environment of the proposed algorithm? Moreover, in the Section Experiment and results, what are the settings of the parameters?

2. The calculation of the evaluation indices in Section 4.2 Evaluation criteria can be deleted due to its simplicity.

3. The reasons why the Authors divided the training set and the test set in a 3:2 ratio should be added. Please add the related references or conduct the parameter setting experiment.

Validity of the findings

1. It seems that the framework of the proposed algorithm is novel. What are the detailed contributions for the proposed algorithm EFS-GINI?

Reviewer 3 ·

Basic reporting

Xin et al present an algorithm which is a three-stage feature selection algorithm framework for high-dimensional data based on ensemble feature selection. Using this algorithm, the authors find the m5c RNA modification. Thanks to the systematic nature of their study and sound methods, the authors are confidently able to describe m5c as a biomarker in kidney cancer across the board, which they associate with reduced survival. The study is overall well rounded on these aspects, both in terms of methods and statistical analysis. The authors also include some further prospects in the regulation of m5c RNA modification. These are the aspects which I think need just a little bit more work to be at the same level as the rest of the paper (see below).

1) Are the methods clear and relicable? Do all the results presented match the methods described?
2) Through bioinformatics analysis, m5C has been identified as a potential tumor biomarker for kidney cancer. How do you plan to validate this finding in subsequent experiments?
3) I observed that the relevant genes are enriched in immune regulation. Immunotherapy has shown high efficacy in the treatment of kidney cancer. Therefore, is it possible that the immune therapeutic effect will be better in patients with high expression of these genes?
4) There are also a few minor quirks regarding the language which are noticed.

Experimental design

no comment

Validity of the findings

no comment

---

## Round 0.2 · accepted · Accept

All reviewers accepted the paper. I recommend acceptance of this paper.

Reviewer 1 ·

Basic reporting

no comment

Experimental design

no comment

Validity of the findings

no comment

Additional comments

no comment

Reviewer 2 ·

Basic reporting

no comment

Experimental design

no comment

Validity of the findings

no comment

Additional comments

I have no further comments; thank you for addressing my comments. The paper and code are now much better than their initial state and ready for publication. Congrats to all authors for this beautiful work.

Reviewer 3 ·

Basic reporting

no comment

Experimental design

no comment

Validity of the findings

no comment